# Hawk Tea Flavonoids as Natural Hepatoprotective Agents Alleviate Acute Liver Damage by Reshaping the Intestinal Microbiota and Modulating the Nrf2 and NF-κB Signaling Pathways

**DOI:** 10.3390/nu14173662

**Published:** 2022-09-05

**Authors:** Ting Xu, Shanshan Hu, Yan Liu, Kang Sun, Liyong Luo, Liang Zeng

**Affiliations:** College of Food Science, Southwest University, Beibei, Chongqing 400715, China

**Keywords:** hawk tea, guaijaverin, antioxidant, inflammatory reaction, alcoholic liver damage, intestinal flora

## Abstract

Hawk tea (*Litsea coreana* Levl. var. *lanuginosa*) is a traditional herbal tea in southwestern China, and was found to possess hepatoprotective effects in our previous study. However, it is unclear whether hawk tea flavonoids (HTF) can alleviate alcoholic liver damage (ALD). Firstly, we extracted and identified the presence of 191 molecules categorized as HTFs, with reynoutrin, avicularin, guaijaverin, cynaroside, and kaempferol-7-O-glucoside being the most prevalent. After taking bioavailability into consideration and conducting comprehensive sorting, the contribution of guaijaverin was the highest (0.016 mg/mice). Then, by daily intragastric administration of HTF (100 mg/kg/day) to the ALD mice, we found that HTF alleviated liver lipid deposition (inhibition of TG, TC, LDL-C) by reducing liver oxidative-stress-mediated inflammation (up-regulation NRF2/HO-1 and down-regulation TLR4/MyD88/NF-κB pathway) and reshaping the gut microbiota (Lactobacillus, Bifidobacterium, Bacillus increased). Overall, we found HTF could be a potential protective natural compound for treating ALD via the gut–liver axis and guaijaverin might be the key substance involved.

## 1. Introduction

Excessive consumption of alcohol is a major cause of liver-related morbidity and mortality. It can lead to alcoholic liver damage (ALD), including fatty liver, fibrosis, and alcoholic hepatitis [1,2,3]. Alcohol abstinence is the most effective therapy for alcoholism and alcohol-related problems; however, withdrawal is not easy, and sudden withdrawal may be associated with protracted alcohol abstinence syndrome [1]. Fortunately, in the past few decades, significant progress has been made in our understanding of the molecular mechanisms underlying alcoholic liver injury [4,5], and treatment based on related signaling pathways has also become an emerging treatment strategy for ALD [6]. Alcoholic liver disease is mainly characterized by oxidative stress, liver inflammation, disturbance of hepatocyte metabolism, and intestinal microbiota dysbiosis [3]. Within the progression of alcoholic liver damage, excessive accumulation of reactive oxygen species (ROS) that are caused by excess alcohol intake is closely linked with the aggravation of ALD through activation of the nuclear factor kappa B (NF-κB) pathway and inactivation of the Nrf2/Keap1 pathway [7]. These two key pathways can not only regulate cellular responses to oxidative stress and inflammation, respectively, but are also involved in many pathological conditions, playing a key role in cancer progression and chemotherapy resistance in many types of cancer [8,9,10,11]. Emerging studies indicate that dysregulated NF-κB and Nrf2/Keap1 pathways cause inflammation-related diseases as well as cancers, and both of them have been long-proposed as potential targets for therapy of diseases [12]. Previous studies reported that alcohol consumption may also lead to intestinal microbiota dysbiosis and further lead to a reduction of beneficial microbial metabolites and the production of harmful microbial metabolites [13]. These factors cause increased gut permeability and bacterial translocation, which stimulate various ALDs [14]. Therefore, a therapeutic strategy that targets the balancing of oxidative-stress-mediated inflammation by modulating specific intracellular signaling pathways or targets the reshaping of microbiomes would provide an approach for the treatment of ALD.

Hawk tea (*Litsea coreana* Levl. var. *lanuginosa*) is a traditional herbal tea in southwestern China. It was documented by the traditional Chinese medicine book “Compendium of Materia Medica” as a hypolipidemic herb, and has been used as a folk medicine for hundreds of years [15]. Modern pharmacological studies have shown the hypolipidemic, hypoglycemic, and anti-inflammatory properties of hawk tea (HT) which were contributed to by its rich bioactive components [15,16]. As a unique plant resource for southwestern China, hawk tea has been found to have the potential to reduce intestinal inflammation and regulate intestinal flora in mice with Dextran Sulfate Sodium (DSS)-induced colitis [17]. Our previous studies indicated that an extract of HT possessed hepatoprotective effects in acute ALD mice, and the phytochemical analysis suggested that flavonoids are the main compounds of HT [18,19]. Thus, HT flavonoids may be a promising liver protectant for ALD. However, the effects of hawk tea flavonoids (HTF) on ALD and their molecular mechanism remain unclear.

This study is aimed at exploring the potential therapeutic effects of HTF on ALD and their molecular mechanism. Here, an ethanol extract of HT was purified using AB-8 macroporous resin to obtain the HTF. The HTF composition was identified using a high-performance liquid chromatography–mass spectrometer/mass spectrometer (HPLC–MS/MS). The therapeutic effects of HTF on acute ALD were studied by testing antioxidant level, liver function index, lipid peroxidation degree, and liver histopathology of the mice. Finally, the underlying molecular mechanism of hepatoprotection from HTF was studied.

## 2. Materials and Methods

### 2.1. Reagents and Chemicals

The HTF was obtained by the method of Cheng et al. with some modifications. 95% ethanol (analytical-grade) was purchased from the Chengdu Chron Chemicals Co., Ltd. (Chengdu, Sichuan, China). The AB-8 macroporous resin was purchased from Yuanye Bio-Technology Co., Ltd. (Shanghai, China). Cytokines in serum and liver (interleukin-1β (IL-1β), IL-6 and tumor necrosis factor-α (TNF-α)) were measured using commercial kits purchased from Nanjing Jiancheng Bioengineering Institute (Nanjing, China). Both serum and liver biochemical indicators (high-density lipoprotein (HDL-C), low-density lipoprotein (LDL-C), triglyceride (TG), total cholesterol (TC), glutamic oxalacetic transaminase (GOT), glutamic pyruvic transaminase (GPT)) were measured using Nanjing Jiancheng commercial kits. Antibodies in the experiment were purchased from Proteintech Group, Inc. (Wuhan, Hubei, China), including the Nrf2 antibody, HO-1 antibody, TLR4 antibody, MyD88 antibody, GAPDH antibody, and Keap1 antibody.

### 2.2. Preparations of HTF

The leaves of hawk tea were harvested in Wuxi County (Chongqing, China) and processed according to the methods of Liu et al. [18]. The HTF was obtained by the method of Cheng et al. with some modifications [20], and the main flow of extraction is shown in Figure 1a. Specifically, the HT was ground in a mortar and standardized using a 40 mesh sieve (0.25 mm). An extraction was made with the HT powder using 95% ethanol (analytical-grade, Chengdu Chron Chemicals Co., Ltd., Sichuan, China) according to a solid/liquid ratio of 1:10 for 4 h, repeated two times. The ethanol extract was concentrated on a rotary evaporator (RV 8, IKA Instrument Equipment Co., Ltd., Guangzhou, Guangdong, China) with a low-temperature cooling-liquid-circulating pump (DLSB-5/20, Changcheng kegongmao Co., Ltd., Zhengzhou, Henan, China) and a water circulation vacuum pump (SHB-3, Changcheng kegongmao Co., Ltd., Henan, China) to obtain the crude HTF.

The crude HTF was further purified and used in subsequent experiments (including Section 2.3, Section 2.4 and Section 2.5). The purification steps were as follows: 10 g/L of crude HTF (HTF before purification) solution was loaded onto an AB-8 macroporous resin (Yuanye Bio-Technology Co., Ltd., Shanghai, China) column (2.6 cm × 60 cm) and eluted by 70% ethanol at an elution flow rate of 5 mL/min. The effluent was collected, concentrated, and freeze-dried to obtained HTF (HTF after purification) which was stored at −80 °C for a maximum of 3 months.

### 2.3. Total Flavonoids Quantification

The assays of total flavonoid content in HTF were carried out using the methodology described by Liu et al. [18]. The total flavonoids of HTF were measured using the aluminum trichloride–sodium nitrite assay. First, 0.05 mL of sample solution was added to 1.35 mL of water followed by 0.05 mL 5% sodium nitrite. After mixing, 0.05 mL 10% aluminum trichloride was added and mixed. Then, the solution was incubated at 25 ± 2 °C for 10 min in the dark, and absorbance values were measured at 415 nm using a Synergy H1MG microplate reader (BioTek Instruments Inc., Winooski, VT, USA). The results were calculated from the rutin standard curve (R^2^ = 0.99 for the calibration curve) and expressed as mg of rutin equivalents (RE)/g.

### 2.4. Qualitation of HTF Composition

The successful application of ultra-high-performance liquid chromatography–tandem mass spectrometry (UHPLC–MS/MS) in the qualitative and quantitative determination of multiple components in complex samples, especially in medical herbs, provides a suitable way to determine different types of compounds in a single run (PMID: 29727783; 31890338). Therefore, we used UHPLC–MS/MS to identify the HTF. The steps are as follows:

HTF was freeze-dried by vacuum freeze-dryer (Scientz-100F). The freeze-dried sample was crushed using a mixer mill (MM 400, Retsch) with a zirconia bead for 1.5 min at 30 Hz. 100 mg of lyophilized powder was dissolved with 1.2 mL 70% methanol solution, vortexed for 30 s every 30 min 6 times in total, then placed in a refrigerator at 4 °C overnight. Following centrifugation at 12,000 rpm for 10 min, the extracts were filtrated (SCAA-104, 0.22 μm pore size; ANPEL, Shanghai, China, http://www.anpel.com.cn/, accessed on 27 January 2021) before UHPLC–MS/MS analysis.

The sample extracts were analyzed using a UHPLC–ESI–MS/MS system (UHPLC, SHIMADZU Nexera X2, www.shimadzu.com.cn/, accessed on 27 January 2021; MS, Applied Biosystems 4500 Q TRAP, www.appliedbiosystems.com.cn/, accessed on 27 January 2021). The analytical conditions were as follows: For UHPLC, the sample was placed in an Agilent SB-C18 column (1.8 µm, 2.1 mm × 100 mm). The mobile phase consisted of solvent A, pure water with 0.1% formic acid, and solvent B, acetonitrile with 0.1% formic acid. Sample measurements were performed with a gradient program that employed the starting conditions of 95% A, 5% B. Within 9 min, a linear gradient to 5% A, 95% B was programmed, and a composition of 5% A, 95% B was kept for 1 min. Subsequently, a composition of 95% A, 5.0% B was adjusted to within 1.10 min and kept for 2.9 min. The flow velocity was set as 0.35 mL per minute; the column oven was set to 40 °C; the injection volume was 4μL. The effluent was alternatively connected to an ESI–triple quadrupole–linear ion trap mass spectrometer (QTRAP–MS). MS was performed in positive and negative modes, and controlled by Analyst 1.6.3 software (AB Sciex).

The ESI source operation parameters were as follows: the ion source was set to turbo spray; source temperature was set at 550 °C; ion spray voltage (IS) was set at 5500 V (positive ion mode)/−4500 V (negative ion mode); ion source gas I (GSI), gas II (GSII), and curtain gas (CUR) were set at 50, 60, and 25.0 psi, respectively; and the collision-activated dissociation (CAD) was high. Instrument tuning and mass calibration were performed with 10 and 100 μmol/L polypropylene glycol solutions in QQQ and LIT modes, respectively. QQQ scans were acquired as MRM experiments with collision gas (nitrogen) set to medium. DP and CE for individual MRM transitions were done with further DP and CE optimization. A specific set of MRM transitions were monitored for each period according to the metabolites eluted within this period.

Based on the self-built database MWDB (Metware Biotechnology Co., Ltd., Wuhan, China) and public databases of metabolite information, the metabolites of HTF were qualitatively and quantitatively analyzed by mass spectrometry. The characteristic ions of each substance were screened out by the triple quadrupole rod, and the signal strength of the characteristic ions were obtained in the detector. The mass spectrometry file under the sample was opened with MultiaQuant software 3.0.3 to carry out the integration and correction of chromatographic peaks, and the relative contents of the corresponding substances in the peak area of each chromatographic peak were calculated. Finally, all chromatographic peak area integral data were derived. Then, we calibrated the mass spectrum peaks based on the information regarding metabolite retention time and peak pattern. After the standard peak of each substance is determined, the area of a single peak is divided by the area of total peak to obtain the relative content of the substance in HTF.

We then obtained the oral bioavailability (OB) and drug similarity (DL) of these substances by searching the literature from 2012 to 2022 using Web of Science, PubMed and TCMSP databases. The metabolite integral values and the corresponding metabolite names of all metabolites detected in this experiment are shown in Appendix A.

### 2.5. In Vitro Antioxidant Activity

The in vitro antioxidant activity of HTF extracts were measured using the 1,1-diphenyl-2-picrylhydrazyl (DPPH) method. The DPPH radical-scavenging capacity was measured using the methodology described by Y. Liu et al. with some modifications, and vitamin C (VC) was used as a positive control and standard [21]. First, 100 μL 0.208 mM DPPH (dissolved in 70% ethanol) was incubated with 100 μL HTF solution (50–2000 μg/mL, dissolved in 70% ethanol) for 40 min at room temperature in the dark. Subsequently, the absorbance at 517 nm was measured using the microplate reader, and loss of DPPH (%) was calculated.

### 2.6. Animal Experimental Design

Thirty male specific-pathogen-free Kunming mice (four-to-five weeks of age, weighing 20 ± 2 g) were obtained from the Laboratory Animal Center at Chongqing Medical University. The animals were kept in a 12 h light/dark cycle SPF environment, and were able to access autoclaved food (Shoobree Xietong Organism Co., Nanjing, China) and distilled water. The air conditions were controlled at a temperature of 18–26 °C and 40–70% relative humidity, and acclimated for 1 week prior to the experiment. All animals were handled according to internationally accepted guidelines, and the protocol was approved by the Experimental Animal Ethical Review Committee of Southwest University (approval No. IACUC-20191123-04).

The acute ALD mice model was established by the method of Yan Liu et al. with some modifications [18]. Mice were randomly divided into five groups as shown in Figure 2a: health group (without flavonoids and alcohol treatments), alcohol group (with alcohol treatment only), HTF-50 group (with 50 mg/kg bodyweight HTF and alcohol treatments), HTF-100 group (with 100 mg/kg bodyweight HTF and alcohol treatments), and HTF-200 group (with 200 mg/kg bodyweight HTF and alcohol treatments). Since the water solubility of flavonoids is poor, all test samples (HTF) were dissolved in 0.5% (*w*/*v*) sodium carboxymethyl cellulose (CMC-Na) solution. First, the mice from HTF-treated groups were intragastrically administered HTF solutions, while the same volume of 0.5% CMC-Na was intragastrically administered to the mice from health and alcohol groups once daily for 5 consecutive days. Then, 50% ethanol (10 mL/kg bodyweight) was intragastrically administered to the mice (including alcohol, HTF-50, HTF-100, and HTF-200 groups) 30 min after HTF treatment once daily for 5 consecutive days, while the same volume of 0.5% CMC-Na was given to the remaining mice.

The bodyweight of mice was measured from the 1st to the 5th day. At the end of the experiment, mice were fasted for 8 h, then anesthetized by intraperitoneal injection of 1% sodium pentobarbital solution before finally being sacrificed using the cervical dislocation method to collect blood and tissues. The blood was collected from anesthetized mice, and was centrifuged at 5000× *g* (Centrifuge-H1850R, Xiangyi Co., Changsha, China) at 4 °C for 10 min to obtain serum. The serum was stored at −80 °C for further analysis. The livers were sampled from sacrificed mice and weighed. Then, parts of livers were used for histopathological analysis and the rest were stored at −80 °C for further biochemical analysis and Western blot analysis.

### 2.7. Biochemical Analysis

The hepatic biochemical indicators (including alcohol dehydrogenase (ADH) activity, acetaldehyde dehydrogenase (ALDH) activity, glutathione (GSH), superoxide dismutase (SOD) activity, malondialdehyde (MDA), total cholesterol (TC), total triglyceride (TG), low-density lipoprotein cholesterol (LDL-C), and high-density lipoprotein cholesterol (HDL-C)) and the serum biochemical indicators (including alanine aminotransferase (ALT), aspartate aminotransferase (AST)) were measured using commercial kits (Nanjing Jiancheng Bioengineering Institute, Nanjing, China). The hepatic biochemical values were normalized to the protein content detected using commercial bicinchoninic acid kits (Nanjing Jiancheng Bioengineering Institute, Nanjing, China) which used bovine serum albumin as a standard.

### 2.8. Histopathological Analysis and Lipid Droplets Quantification

The liver tissue was collected and immediately fixed in 10% natural formalin, embedded in paraffin, cut into 5 μm-thick pieces, and stained with hematoxylin and eosin (H&E). Then, all the liver slices were analyzed under a light microscope (Nikon-E100, NIKON, Tokyo, Japan). For the Oil Red O staining, hepatic sections were frozen in liquid nitrogen, sliced, and stained with Oil Red O solution (0.5 g/100 mL, dissolved in isopropanol). The area and number of lipid droplets were calculated using the Fiji (NIH) particle analyzer tool [22].

### 2.9. ELISA Kit Analysis

Tumor necrosis factor α (TNF-α), interleukin-6 (IL-6), and interleukin-1β (IL-1β) levels in the mouse liver were determined using the corresponding ELISA kits following the manufacturers’ instructions (Nanjing Jiancheng Bioengineering Institute, Nanjing, China), and the product code is shown in Appendix A. The results were normalized to the total protein.

### 2.10. Western Blot Analysis

The process of Western blot analysis refers to previous research methods of our laboratory [23]. The liver tissue of mice was washed with cold phosphate buffer solution (PBS) and then treated with a RIPA buffer (RIPA; Servicebio, Wuhan, China) containing a protease inhibitor cocktail. The proteins were separated by 10% SDS/PAGE and transferred to a PVDF membrane. After blocking in 5% skimmed milk for 1 h, the PVDF membrane was incubated overnight with primary antibodies at 4 °C and then with secondary antibodies at room temperature for another 1 h. An enhanced chemiluminescence light (ECL) detection kit (ECL, Servicebio, Wuhan, China) was used for protein detection, and AlphaEase FC software was used to determine the value. Antibodies in the experiment were purchased from Proteintech Group, Inc. (Wuhan, Hubei, China), including the Nrf2 antibody, HO-1 antibody, TLR4 antibody, MyD88 antibody, GAPDH antibody, and Keap1 antibody (Product code is shown in Appendix A).

### 2.11. Gut Microbiota Analysis

Fecal microbial DNA was extracted according to the instructions provided in the extraction kit (Omega Bio-Tek, Norcross, GA, USA), and the V3–V4 region of the 16s rRNA gene was amplified by PCR (forward primer (5′-ACTCCTACGGGAGGCAGCAG-3′) and reverse primer (5′-GGACTACHVGGGTWTCTAAT-3′)). The PCR products were purified and quantified using the DNA gel extraction kit (Axygen Biosciences, Union City, CA, USA) and a Quantus Fluorometer (Promega, Madison, WI, USA). Sequencing was completed using Illumina’s NovaSeq PE300 platform (Majorbio Bio-Pharm Technology Co., Ltd., Shanghai, China). Fastp (version 0.19.6, https://github.com/OpenGene/fastp, accessed on 9 January 2022) and FLASH (version 1.2.11, https://ccb.jhu.edu/software/FLASH/index.shtml, accessed on 9 January 2022) software programs were used for quality control and splicing. The DADA2 plug-in for Qiime2 (version 2020.2, https://qiime2.org, accessed on 16 January 2022) was used for denoise sequences. Data were analyzed based on the Majorbio Cloud Platform (www.majorbio.com, accessed on 10 June 2022).

### 2.12. Statistical Analysis

GraphPad Prism 8.0 (GraphPad Software, San Diego, CA, USA) was used for statistical analysis. All data were presented as Mean ± SEM. Duncan’s test was used to compare among groups by analysis of variance (ANOVA); *p* < 0.05 indicates a statistically significant difference and *p* < 0.01 indicates a highly statistically significant difference. Graphs were obtained using GraphPad Prism 8.1 and OriginPro, 2019b (OriginLab Corporation, Northampton, MA, USA).

## 3. Results

### 3.1. In Vitro Antioxidant Activity and Identification of HTF

By extracting for purification, we obtained HTF with a purity of 95.6%. We used the DPPH method to evaluate the in vitro antioxidant activity of HTF, and found that purification could improve the DPPH radical-scavenging ability of HTF, and that the antioxidant activity in vitro was positively correlated with the concentration of HTF solution (Figure 1b). We identified and characterized the presence of 191 molecules belonging to the flavonoid family. The result of the HPLC–MS/MS analysis of flavonoid compounds present in the freeze-dried hawk tea is shown in Appendix A.

The flavonoids can be subdivided into 11 different subgroups depending on which carbon of the C ring the B ring is attached to as well as the degree of unsaturation and oxidation of the C ring. Flavonols, flavones, and flavonoid carbonosides were the major subgroups of HTF, accounting for more than 80% of HTF (Figure 1c). Then, we obtained the oral bioavailability (OB) and drug similarity (DL) of these substances by searching the literature from 2012 to 2022 using Web of Science, PubMed, and TCMSP databases (Appendix A). Among them, flavonols (guaijaverin and quercetin), flavones (apigenin and luteolin), flavonoid carbonosides (orientin and vitexin), flavanols (epicatechin and catechins), chalcones, and isoflavones (genistein and daidzein) were found to be compounds present in great amounts and in high biological activities.

Then, we calculated the equivalent value of the monomer component of HTF as its contribution at a daily oral dose of 100 mg/kg in mice. The calculation formula is as follows:Contribution=Dose of HTF×Concent in HTF×Oral bioavilability.

After screening and statistical analysis of substances with a proportion of more than 1% of the HTF content and substances with high biological utilization (OB ≥ 30% or DL ≥ 0.18), 25 types of flavonoid monomers were sorted (as shown in Table 1). Particularly, we found that guaijaverin, which ranks first in contribution (0.016 mg/mice) and has twice that of trifolin (ranks the second, 0.008 mg/mice), is most likely to be the key effective substance to protect from ALD after oral administration.

### 3.2. HTF Reduced Alcohol-Induced Body Weight Loss

All the mice had similar initial body weight (Figure 2a,b). However, the intake of alcohol significantly affected body weight gain; it led to a weight loss of about 14.2%. HTF significantly reversed this trend (Figure 2b).

### 3.3. HTF Reduced Alcohol-Induced Oxidative Stress

Acute alcohol intake may influence the oxidation–antioxidant balance of the body, and effects include reducing the activity of alcohol metabolic enzymes in the liver, which can destroy the antioxidant system and down-regulate the antioxidant pathway. In our study, we found HTF can improve the metabolic rate of alcohol by increasing the activity of alcohol metabolic enzymes, in which the activity of ADH and ALDH is increased by 222.2% (Figure 2c) and 108.3% (Figure 2d) upon the administration of HTF-200, respectively. Moreover, the administration of HTF at three doses (50, 100, and 200 mg/kg BodyWight) reduced the inhibitive effect of alcohol intake on the internal antioxidant system (Figure 2e–g). The 200 mg/kg HTF dosage improved the content of antioxidants (GSH) by 26.9% and the activity of antioxidases (SOD) by 42.1%. In addition, HTF inhibited TLR4, MyD88, and NF-κB levels. We also found that HTF-intervention mice had higher expression of Nrf2 and HO-1 proteins (Figure 2k–m) compared to the alcohol group.

### 3.4. HTF Inhibited Hepatic Inflammation in Acute ALD-Model Mice

Alcohol intake up-regulated the expression of inflammatory pathway proteins, increased the expression of inflammatory factors, and aggravated inflammation. However, it was observed that HTF administration markedly inhibited the inflammatory response in the liver of ALD mice, which was accompanied by a significant inhibition of serum pro-inflammatory cytokines, such as IL-6, IL-1β, and TNF-α (Figure 2h–j). In particular, HTF-200 performed better than the other two lower doses and significantly reduced IL-6 expression in the liver. In addition, HTF inhibited TLR4, MyD88, and NF-κB levels in a concentration-dependent pattern (Figure 2l,m). Notably, HTF-200 decreased the expression of TLR4 by 40.8% and the activity of p65 by 57.8%.

### 3.5. HTF Reduced Alcohol-Induced Lipid Deposition in the Liver

Alcohol intake may also lead to lipid deposition in the liver. However, in our study, it was found that HTF can reduce alcohol-induced liver lipid deposition. It is mainly manifested in the decrease of biochemical indexes of liver lipid TC, TG, and LDL-C (Figure 3a–d). Respectively, the hepatic TC, TG, and LDL-C levels in the HTF-200 administrated mice were lower by a 0.29-fold, 0.45-fold, and 0.58-fold factor than that of the alcohol-drinking mice. These results were also confirmed by HE staining and Oil Red O staining of the liver. As shown in Figure 3h, local inflammation and necrosis occurred in liver cells of the alcohol-drinking mice, and their cytoplasm became loose and empty while nuclear cells were suspended in the center and became shallow in staining. In addition, Oil Red O staining of liver sections showed enlarged lipid droplet formation, preferentially in the alcohol-treated liver. Some hepatocytes had undergone steatosis, and aggregated lipid droplets could be seen in their interior. The local inflammation and extensive lipid droplet formation were reduced in liver cells of the HTF-administered mice. As shown in Figure 3i, compared with the alcohol group, the mice treated with HTF demonstrated reduced cytoplasmic vacuolation of liver cells, and the lipid droplets in liver cells were small and scattered. In liver cells, all doses of HTF induced a decrease in the percentage of cytoplasm area occupied by lipid drops, from 47.65% in the alcohol group to 42.97%, 32.54%, and 23.34% in the HTF-50, HTF-100, and HTF-200 treatment groups, respectively (Figure 3g).

### 3.6. HTF Reshaped the Gut Microbiota in Acute ALD-Model Mice

Alcohol intake may also lead to gut microbiota change. HTF may have reshaped the intestinal flora in the alcohol group. To explore how alcohol affects gut microbiota and the role of HTF in the regulation of gut microbiota in ALD mice, 16S rRNA sequencing was used to analyze the composition of gut microbiota. The Ace index and the chao index at the ASV level for total bacterial community diversity were both increased during alcohol treatment (Figure 4a and Appendix A). Based on the weighted UniFrac Principal Coordinate Analysis (PCoA) diagram, the microbial community composition from different groups showed obvious clustering (Figure 4b,c). Based on the phylum level, HTF significantly increased the relative abundance of Firmicutes and decreased the abundance of Bacteroidota. The Firmicutes/Bacteroidota (F/B) ratio was also reduced by alcohol, while HTF-100 and HTF-200 significantly reversed this trend (Figure 4d,e and Appendix A). At the genus level, HTF restored the abundance levels of Lactobacillus, which had been reduced by alcohol intake (Figure 4f,g and Appendix A). The results of analyzing microbial composition for the HTF and alcohol groups showed that HTF could reverse the alterations of gut microbiota in ALD mice by increasing the abundance of *Lactobacillus*, *Bifidobacterium*, *Bacillus* and decreasing the abundance of *Bacteroides*, *Prevotellaceae_NK3B31_group*, *Alloprevotella*, *Prevotellaceae_UCG-001*, *norank_o__Clostridia_UCG-014*. (Appendix A).

Based on the results of relative abundances, heatmap analysis was carried out for the top 50 abundant genera (Figure 5). Figure 5a shows significant differences in the gut microbiota composition between the health groups and the alcohol groups at the genus level. The color was defined at the phylum level. The results showed that the relative abundance of *Lactobacillus* was positively correlated with that of the HTF treatment group. Next, we investigated the correlation between gut microbiota and metabolic factors (Figure 5b). Some metabolic factors (TC, TG, etc.) and inflammatory factors (IL1β, IL6, TNF-α) were negatively correlated with *Lactobacillus*, *Bifidobacterium* and *Bacillus*, while positively correlated with *Prevotellaceae_NK3B31_group*, *Prevotellaceae_UCG-001* and *norank_o__Clostridia_UCG-014.*

## 4. Discussion

After alcohol enters the body, about 80% of the alcohol is transformed in the liver, wherein it is metabolized into other toxic compounds (such as the highly toxic intermediate metabolite, acetaldehyde). The direct effects of ethanol metabolism include disruption of internal oxidative/antioxidant balance, resulting in the accumulation of ROS and activation of the NF-κB pathway. This activation causes endoplasmic reticulum stress, induces a cascade of inflammation, and finally results in hepatic lipid metabolism disorders, which have been recognized as important events in the development of ALD. The binding of acetaldehyde to unfolded proteins impairs lipoprotein secretion, which can also significantly enhance liver lipid accumulation [3]. In addition, with the discovery of the gut–liver axis, it has been suggested that alcohol-mediated intestinal-environmental damage and microbial dysregulation may also be one of the important pathogeneses of alcoholic liver injury [14,51].

Currently, plant and herb extracts are applied toward the treatment of chronic and acute conditions and various ailments and problems such as metabolic syndrome, inflammation, cancer, etc., because they act as extensive sources of rich phytochemical components [52,53,54,55]. It is estimated that about 25% of the drugs prescribed worldwide are derived from plants, and that 121 such active compounds are in use [56]. Therefore, it is necessary to explore the protective mechanisms of natural phytochemicals against alcoholic liver injury. As a traditional herb in southwest China, hawk tea is rich in flavonoids, polyphenols, polysaccharides and other bioactive substances, and showed significant liver protective, anti-inflammatory, antioxidative, anti-obesity, and other biological activities through a variety of molecular mechanisms [15]. The flavonoids are the main bioactive substances of hawk tea. In our research, the results of LC–MS indicated that HTF contains several flavonoids, which can be divided into 11 diverse subclasses. Among them, flavonols (guaijaverin and quercetin), flavones (apigenin and luteolin), flavonoid carbonosides (orientin and vitexin), flavanols (epicatechin and catechins), chalcones, and isoflavones (genistein and daidzein) are classes of compounds present in great amounts and in high biological activities.

Further, the monomer composition of HTF was analyzed. In order of content, the compounds with the highest content were reynoutrin (5.06%), avicularin (3.47%), guaijaverin (2.72%), cynaroside (2.60%), and kaempferol-7-O-glucoside (2.21%). According to the relevant literature, we found that substances with high concentrations also have high antioxidant and anti-inflammatory activities; one such substance, avicularin, was found to alleviate chemical liver injury due to its antioxidant properties. HTF is a mixture of flavonoids that is metabolized into small molecules in the small intestine after oral administration, after which it enters the liver to generate effects useful for therapeutic intervention. Therefore, biological utilization of the monomer is particularly important [55], in addition to its proportional content within HTF. After screening and statistical analysis of substances with a proportion of more than 1% of the HTF content and substances with high biological utilization (OB ≥ 30% or DL ≥ 0.18), 25 types of flavonoid monomers were sorted (as shown in Table 1). Then, we calculated the contribution of these substances and made a comprehensive ranking. Particularly, we found that guaijaverin, which ranks first in contribution (0.016 mg/mice) and 100.0% higher than trifolin (the second, 0.008 mg/mice), is most likely to be a key effective substance to protect from ALD after oral administration. There are also studies that show the anti-inflammatory and antioxidant activity of guaijaverin, as well as its demonstrably high antibacterial activity against Strep. mutans. [24,25]. We also found that the glycoside compounds occupies a large proportion of the HTF, and possess their own high degree of antioxidative, anti-inflammatory, and immune-regulatory activity. So, the correlated glycoside may also have a potential effect in regards to the relief of liver injury [28,31,49]. However, the evaluations of these chemical components still need to be further explored, due to the lack of their biological utilization data.

Our study shows that HTF may alleviate alcohol-induced oxidative stress by increasing the activity of alcohol metabolism enzymes in the liver, maintaining antioxidant system activity, and up-regulating the expression of oxidative stress pathway proteins in vivo. HTF can improve the efficiency of alcohol metabolism by increasing the liver ADH and ADLH activity induced by acute alcohol exposure. In particular, HTF-200 intake increased ADH and ALDH activities by 2.22 and 1.08 times, respectively. As the main enzymes of the ethanol-metabolic pathway, the enzymes alcohol dehydrogenase (ADH) and acetaldehyde dehydrogenase (ALDH) convert ethanol into acetate via the intermediate acetaldehyde [57,58]. When the activity of ADH and ALDH decreases, the metabolic rate of alcohol decreases, leading to the accumulation of reactive oxygen species (ROS)—mainly hydrogen peroxide (H_2_O_2_) and superoxide anion O^2−^—which cause oxidative stress, ER stress, and steatosis [3,59]. HTF is able to improve the activity of the internal antioxidant system (enzymatic and non-enzymatic) and reduce lipid peroxidation resulting from acute alcohol intake. Alcohol-consumption-induced oxidative stress can lead to excessive ROS accumulation and MDA formation, which can be reduced by antioxidant enzymes SOD and GSH, protecting various cells against oxidative damage [4,60]. In our experiments, HTF treatment dramatically induced an increase in SOD activities and GSH contents and caused a decrease in the levels of MDA in alcohol-induced liver damage mice. The HTF also showed stronger DPPH radical-scavenging activity in vitro. Taken together, these investigations implied that HTF is able to improve the activity of the internal antioxidant system (enzymatic and non-enzymatic) from acute alcohol intake and help to keep oxidant/antioxidant balance.

The enhanced levels of internal antioxidant systems might be attributable to the upregulation of the Nrf2 pathway [61]. Nrf2 is a major regulator of the antioxidant stress response and is usually confined to the cytoplasm by Keap1. Nrf2 is activated when cells are subjected to oxidative or chemical stress (such as alcohol). The activated Nrf2 is then released from Keap1 and translocated into the nucleus, whereupon it binds to AREs and regulates the expression of antioxidant stress genes and enzymes including NADPH, NAD(P)H, glutathione peroxidase (GPx), and heme oxygenase-1 (HO-1) [62]. Our results demonstrate that HTF apparently induced a decrease of Keap1 and an increase of Nrf2 in total liver homogenate, which were related to enhancing the nuclear translocation of Nrf2 and heightening ARE luciferase activity in a dose-dependent manner in the liver. Additionally, our findings indicated that various dosages of HTF treatment effectively enhanced expressions of HO-1 in the liver. In this study, we found HTF treatment could effectively up-regulate Nrf2 and HO-1 protein expression and down-regulate Keap1 protein expression, preliminarily validating the involvement of the Keap1/Nrf2/HO-1 pathway in the protection offered by HTF against damage in the ALD mice.

In fact, as a highly active plant-metabolic product, many flavonoids have been reported as antioxidants. It has been reported by a number of researchers that there exists a strong antioxidant effect with flavonoids such as guaijaverin, myricetin-3-O-glucoside, fisetin, and nicotiflorin (Table 1). In addition, there are flavonoids such as myricetin, avicularin, quercetrin, and rutin which help to inhibit the production of superoxide radicals (Table 1). Sang Min Kim [34] suggested that avicularin and quercetin have clear hepatoprotective activity against injury by t-BHP in HepG2 cells, and the hepatoprotective effect of them showed a high correlation with radical-scavenging activity [34]. Research has also shown that flavonoids are very effective in preventing lipid peroxidation [63], which is supported by our study. In several animal models, citrus flavonoids like hesperidin has been demonstrated to increase GSH content, enhance antioxidant enzyme activities (SOD, CAT, and GPx), and mediate the expression of Nrf2 and up-regulation of antioxidant status [64]. Therefore, in this study, some monomers of HTF may relieve oxidative stress in ALD mice by maintaining the activity of the antioxidant system in vivo, reducing lipid peroxidation, and up-regulating the expression of oxidative stress pathway proteins in vivo, independently or synergistically.

Oxidative stress may contribute to an increase in inflammation during alcohol metabolism. Increased inflammation further exacerbates liver tissue necrosis, forming an inflammatory cascade with oxidative stress. HTF can reduce inflammation caused by alcohol, down-regulate the expression of inflammatory pathways, and decrease pro-inflammatory cytokine secretion in the liver. Alcohol intake may lead to increased intestinal LPS concentration which could be translocated to the liver through blood circulation, thereby activating the TLR4/MyD88/NF-κB pathway and inducing pro-inflammatory cytokine secretion in macrophages (Kupffer cell) [59]. Fascinatingly, HTF prominently protected against alcohol-induced liver damage in mice, which was accompanied by a significant inhibition of serum pro-inflammatory cytokines such as TNF-α and IL-1β. A number of contributing factors, such as alcohol itself, its toxic metabolites, and over-expression of ROS can induce inflammation in liver tissues. As a result, TLR4/MyD88/NF-κB is activated and further, different inflammatory factors are released, including TNF-α, TNF-1β, IL-6, etc. In fact, our results have shown that alcohol enhanced TLR4 and MyD88 levels and over-expression of pro-inflammatory factors. These factors, in conjunction with oxidative stress, combined with ROS accumulation induced by alcohol intake, may lead to increased ER stress and further activation of the NF-κB pathway, resulting in a cascade of inflammation [59]. HTF markedly suppressed the protein expression of TLR4 and MyD88 and down-regulated the NF-κB metabolic pathway in a concentration-dependent pattern, avoiding the inflammatory cascade induced by activation of this pathway. As indicated previously, excessive ROS can exacerbate the inflammatory response and promote the expression of pro-inflammatory cytokines which further increase the intensity of oxidative stress, resulting in a vicious cycle of oxidative stress and inflammation [65]. This may be due to the strong correlation between flavonoids and the inhibition of the NF-κB-related mechanisms [66]. Studies have indicated the great effects of flavonoids on a variety of inflammatory processes and immune functions. It has been shown that various flavonoids such as reynoutrin, guaijaverin, cymaroside, quercetin, kaempferol, and other flavonoids not only help inhibit the initial process of inflammation by decreasing the inflammatory markers such as IL-6, IL-1β, TNF-α, thus regulating the gene expression of anti-inflammatory genes, but also contribute to improving the immune system (Table 1). Comalada et al. [67] reviewed the effects of flavonoids, particularly quercetin, on a variety of inflammatory processes and immune functions and found that certain flavonoids help inhibit the initial process of inflammation and improve the immune system. Because of this, while explaining the possible mechanism of flavonoids in regarding to effective liver injury prevention, we further speculate that HTFs have complementary and overlapping mechanisms of action; these include antioxidant activity, scavenging of free radicals, modulation of enzyme activities in detoxification, oxidation, and reduction, regulation of pro-inflammatory genes, inhibition of inflammatory cascades, anti-inflammatory properties, and action on other possible targets.

In addition, increasing evidence has confirmed that increased inflammation and intensity of oxidative stress aggravate the accumulation of lipids in the liver [3,68]. Therefore, we hypothesize that oxidative stress and inflammation may be two important factors leading to lipid accumulation in the liver, and that the vicious cycle of oxidative stress and inflammation is likely to aggravate hepatic lipid metabolism disorders. In our study, we observed that HTF reduced alcohol-induced lipid deposition in the liver. The hepatic TC, TG, and LDL-C decreased in the HTF-ingested mice, which was also confirmed by H&E staining and Oil Red O staining histological sections of the liver. In the Oil Red O staining slice, we observed a similar result, that is, local lipid droplets in the hepatocytes of mice after HTF administration had been lowered. In liver cells, all doses of HTF induced a decrease in the percentage of cytoplasm area occupied by lipid drops, from 47.65% in the alcohol group to 42.97%, 32.54%, and 23.34% in the HTF-50, HTF-100, and HTF-200 treatment groups, respectively. Our results confirm that HTF administrated to ALD mice decreased fat storage in adipocyte-like tissues. Lipid drop size and total area occupied by lipid droplets were decreased after HTF administration. This finding is in accordance with a previous study reporting that flavonoids such as genistein (an isoflavone), xanthohumol (a chalcone), and isoquercetin (a flavonol) suppressed lipid droplet accumulation on larval tissues of fruit fly fat bodies and thereby reduce liver lipid deposition [22]. How does HTF reduce liver lipid deposition caused by alcohol intake? In our study, the regulation of oxidative stress and inflammation by HTF may be an important indirect pathway in the suppression of ALD-related lipid peroxidation. Several reviews on flavonoids have also mentioned that flavonoids, including citrus flavonoids (such as tangerine peel, naringenin) as well as tea flavonoids, can improve non-alcoholic fatty liver induced by high fructose intake, Western diet, etc. Their potential mechanisms include the fact that flavonoids may affect classical oxidative stress and inflammatory pathways, thereby reducing two important factors (inflammation and oxidative stress) that accelerate disease progression [64,69,70]. Combined with what we mentioned above, oxidative stress and inflammation may aggravate the accumulation of lipids in the body. In addition, flavonoids and their corresponding glycosides can directly prevent lipid accumulation by affecting adipogenesis or accelerating lipid release. Among them, flavonoids such as genistein, naringenin, luteolin, rutin, etc. have been shown to significantly down-regulate SREBP-1, FAS, SCD, and ACC related to de novo fat synthesis, thereby reducing hepatic lipogenesis [71,72,73]. In addition, flavonoids (such as myricetin, nobiletin etc.) also increased the expression of CPT enzymes and genes such as PPAR-α and PPAR-γ, which are all involved in fatty acid oxidation [74,75]. As indicated previously, HTF may influence alcohol-induced liver lipid deposition through the indirect means of alleviating oxidative stress and inflammation, as well as directly though regulation of metabolic pathways related to lipid production and output [75].

Liver inflammation and hepatic steatosis may also be associated with gut dysbiosis in the alcohol-treated mice [68,76]. Studies have shown that alterations in the type and amount of microorganisms are important elements in the ALD-related pathogenesis [51,59,77]. In fact, ALD can cause intestinal flora disorders, which means quantitative (bacterial overgrowth) and qualitative (dysbiosis) changes in the intestinal microflora [51]. Our research shows that HTF can reshape the intestinal flora disorder caused by alcohol and prevented the F/B ratio to that in healthy mice. Moreover, at the genus level. We also observed that *Lactobacillus* abundance was decreased in the alcohol group, but significantly increased in HTF-100 and HTF-200 mice. *Lactobacillus*, as a common symbiotic bacterium in the gut [78], produces lactic acid, which maintains a relatively low pH in the gut and inhibits the growth of pathogens [79]. HTF-100 was also found to increase the abundance of Bifidobacterium and *Bacillus*. Related analysis found that *Lactobacillus* and *Bifidobacterium* are significantly negatively related to MDA, while *Bifidobacterium* and *Bacillus* are significantly positively related to SOD. Interestingly, the presence of *Lactobacillus* and *Bifidobacteria* is associated with healthy intestinal mucosa and strong host immune function [51,80,81]. Both of them are thought to help restore/maintain intestinal flora balance in excessive-drinking patients and thus effectively improve ALD [51]. These results suggest that HTF may play an important role in reshaping the intestinal flora in ALD mice. In addition, our study also found that HTF reduced alcohol-induced excessive growth of *Bacteroides*, *Prevotellaceae_NK3B31_group*, *Alloprevotella*, *Prevotellaceae_UCG-001*, *norank_o__Clostridia_UCG-014*. In some cases of liver injury, *Bacteroides* and *Clostridium* are markedly elevated, producing large amounts of secondary bile acids and triggering an inflammatory response, causing liver cell damage which could lead to a deterioration of liver disease [82,83]. So, comprehensively, alcohol-induced gut dysbiosis can be rebalanced by HTF treatment. These results suggest that HTF may play a potential role in curing alcohol-related liver diseases caused by acute alcohol consumption.

Our experimental results showed that the intestinal flora of ALD-model mice were broken and promoted inflammatory factors. Inflammatory factors and the bacteria invaded the liver through enterohepatic circulation, and then aggravated the hepatic inflammatory and lipid metabolism disorders, before finally accelerating fatty liver and liver fibrosis [84]. Increased TLR-4 and pro-inflammatory cytokines (IL1β, IL6, TNF-α) levels were detected in ALD-model mice; this was associated with bacterial overgrowth or microbiota changes as well as increased intestinal permeability [3]. It seems a there is a possible involvement of the gut–liver axis through the LPS/TLR-4/pro-inflammatory cytokines axis in the pathogenesis of inflammatory processes. Our lab’s early studies have shown that the reduction of intestinal inflammation and liver toxicity by tea (*Camella sinensis)* and hawk tea (*Litsea coreana*) is due to the reshaping of gut microbes [17,23,54,84], which further supported the finding reported in this paper that the ALD-mediated intestinal microorganism composition affects intestinal and liver inflammation. The intestinal-flora-disorder-mediated activation of TLR4 signaling pathways aggravates lipid metabolic disorders, further causing liver lipid deposition.

In this article, we first engage in extraction, isolation, purification, and component identification of HTF; 191 components are identified. Among them, reynoutrin (5.06%), avicularin (3.47%), guaijaverin (2.72%), cynaroside (2.60%), and kaempferol-7-O-glucoside (2.21%) were the most abundant compounds belonging to the category of HTFs. Moreover, they were reported to possess their own high antioxidant and anti-inflammatory activity. After consulting the literature, we found that bioavailability has a great influence on its bioactivity in vivo. Therefore, we screened for substances with a content ratio of more than 1% and higher bioavailability (OB ≥ 30% or DL ≥ 0.18), and ranked their contribution by an integrated calculation. We found that guaijaverin’s contribution (0.016 mg/mice) is the highest, and twice as high as trifolin’s (the second highest, 0.008 mg/mice). Guaijaverin’s antioxidant and anti-inflammatory activities have been reported in the literature, and they may play an important role in alleviating alcoholic liver injury.

## 5. Conclusions

Our results indicated that HTF (100 mg/kg/day) can alleviate lipid metabolism (inhibition of TG, TC, LDL-C) by reducing liver oxidative stress-mediated inflammation (up-regulation NRF2/HO-1 and down-regulation TLR4/MyD88/NF-κB pathway) and reshaping the gut microbiota (*Lactobacillus*, *Bifidobacterium*, *Bacillus increased*), thereby protecting against alcohol-induced liver damage. Overall, HTF could be a potential protective natural compound for ALD via the gut–liver axis and guaijaverin might be the key substance involved. We also observed that, in our study, some substances with higher content in HTF (such as cynaroside, quercetin-3-O-(6″-acetyl) galactoside, quercetin-3-O-robinobioside, etc.) may alleviate alcoholic liver injury due to their high antioxidant and anti-inflammatory activity according to literature reports. However, we found no data regarding their bioavailability so far by searching the literature from 2012 to 2022 using Web of Science, PubMed and TCMSP databases [85]. Therefore, the bioavailability of these substances can be evaluated in subsequent experiments to facilitate their potential functional development. Moreover, in our experiments, HTF was observed to significantly improve hepatic lipid deposition (reduce TG accumulation and promote lipid droplet reduction). It will be our next research goal to further explore how the HTF inhibits the hepatic lipid accumulation (lipid droplet fusion) during the development of alcoholic fatty liver.

## Figures and Tables

**Figure 1 nutrients-14-03662-f001:**
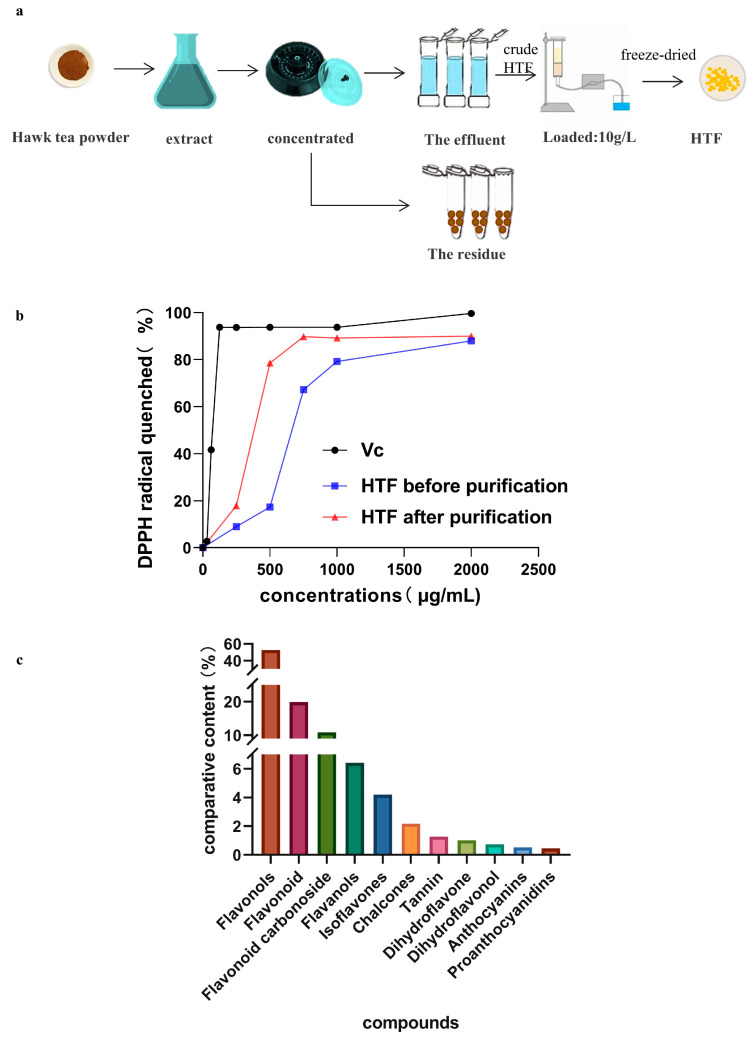
Flavonoids in hawk tea. (**a**) Extraction method of hawk tea flavonoids. (**b**) The DPPH radical quenched percentage of HTF before and after purification. (**c**) Qualification of HTF composition. HTF: hawk tea flavonoids; VC: vitamin C.

**Figure 2 nutrients-14-03662-f002:**
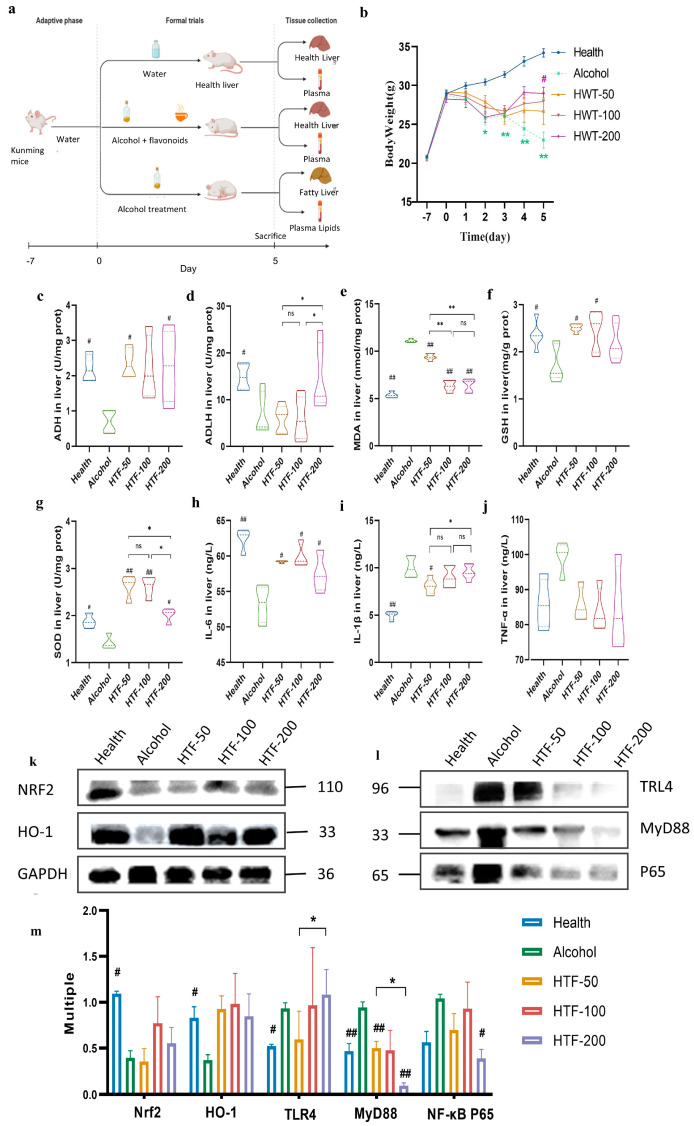
HTF reduced alcohol-induced oxidative stress and inflammatory response in acute ALD-model mice. (**a**) The experimental group. (**b**)Body weight. (**c**) Level of ADH. (**d**) Level of ALDH. (**e**) Level of MDA. (**f**) Level of GSH. (**g**) Level of SOD. (**h**) Level of interleukin-6 (IL-6). (**i**) Level of interleukin-1(IL-1β). (**j**) Level of TNF-α. (**k**) Western blot. (**l**) Western blot. (**m**) Western blot analysis (*n* = 3 per group). Data are mean ± SEM. Significance was determined by one-way ANOVA corrected for multiple comparisons with Dunnett’s test. # *p* < 0.05 and ## *p* < 0.01 compared with Alcohol group. ns, *p* > 0.05, * *p* < 0.05 and ** *p* < 0.01 compared with other HTF group.

**Figure 3 nutrients-14-03662-f003:**
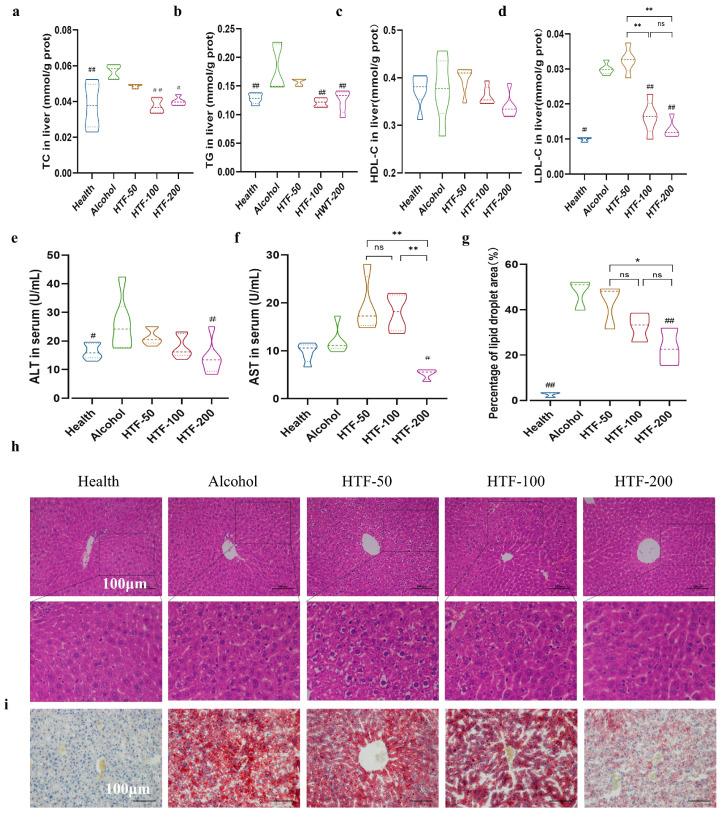
HTF reduced alcohol-induced lipid deposition in the liver. (**a**) Hepatic TC level. (**b**) Hepatic TG level. (**c**) Hepatic LDL-C level. (**d**) Hepatic HDL-C level. (**e**) Serum ALT level. (**f**) Serum AST level. (**g**) Percentage of lipid droplet area. (**h**) Histological examination of liver tissues. (**i**) The Oil Red O staining slice. Data are mean ± SEM. Significance was determined by one-way ANOVA corrected for multiple comparisons with Dunnett’s test. # *p* < 0.05 and ## *p* < 0.01 compared with alcohol group. ns, *p* > 0.05, * *p* < 0.05 and ** *p* < 0.01 compared with other HTF group.

**Figure 4 nutrients-14-03662-f004:**
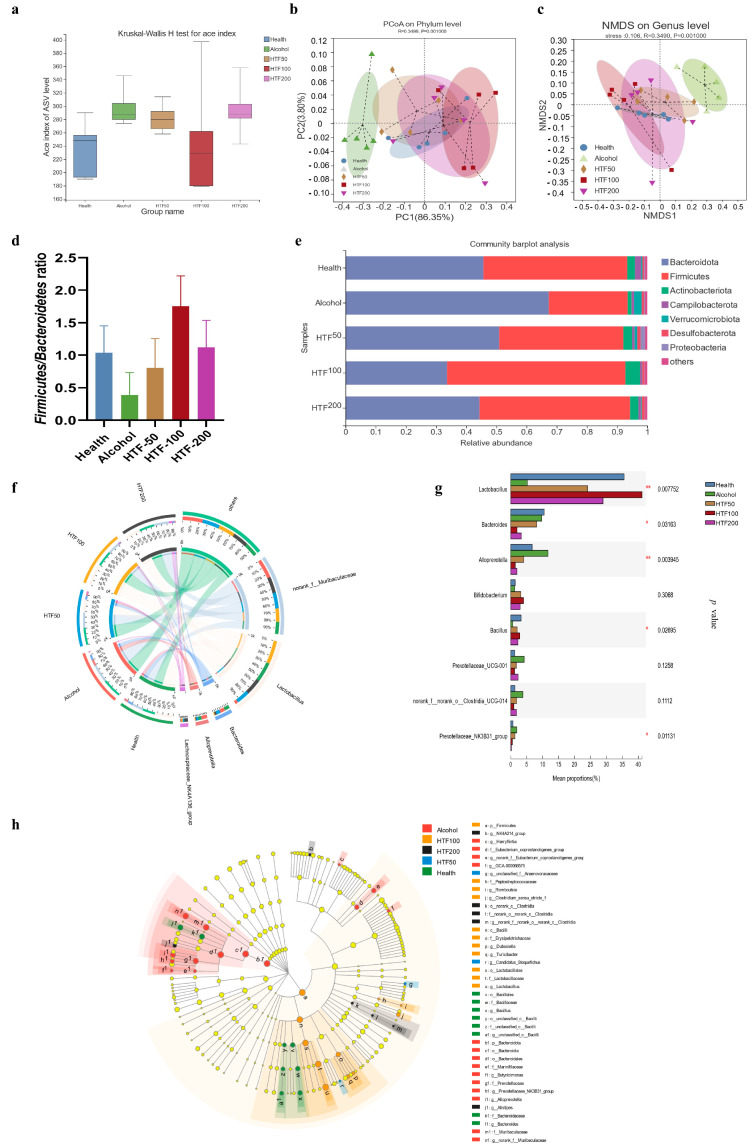
HTF reshaped the gut microbiota in alcohol-induced mice. (**a**) α-Diversity representing the Ace index at the ASV level. (**b**) PCoA plot of the gut microbiota based on weighted UniFrac matrixes. (**c**) Typing analysis of the phylum level in different treatment groups. (**d**) The Firmicutes/Bacteroidetes ratio of all groups. (**e**) Analysis of gut microbiota composition at the phylum level. (**f**) Relative abundance of the fecal microbiota with relative abundance greater than 1% at the phylum level. (**g**) Analysis of gut microbiota composition at the genus level. (**h**) Indicator bacteria of intestinal microbiota with LDA scores of 2 or greater in mice with different treatments. Data are mean ± SEM (*n* = 5 per group). * *p* < 0.05, ** *p* < 0.01 vs. the alcohol group.

**Figure 5 nutrients-14-03662-f005:**
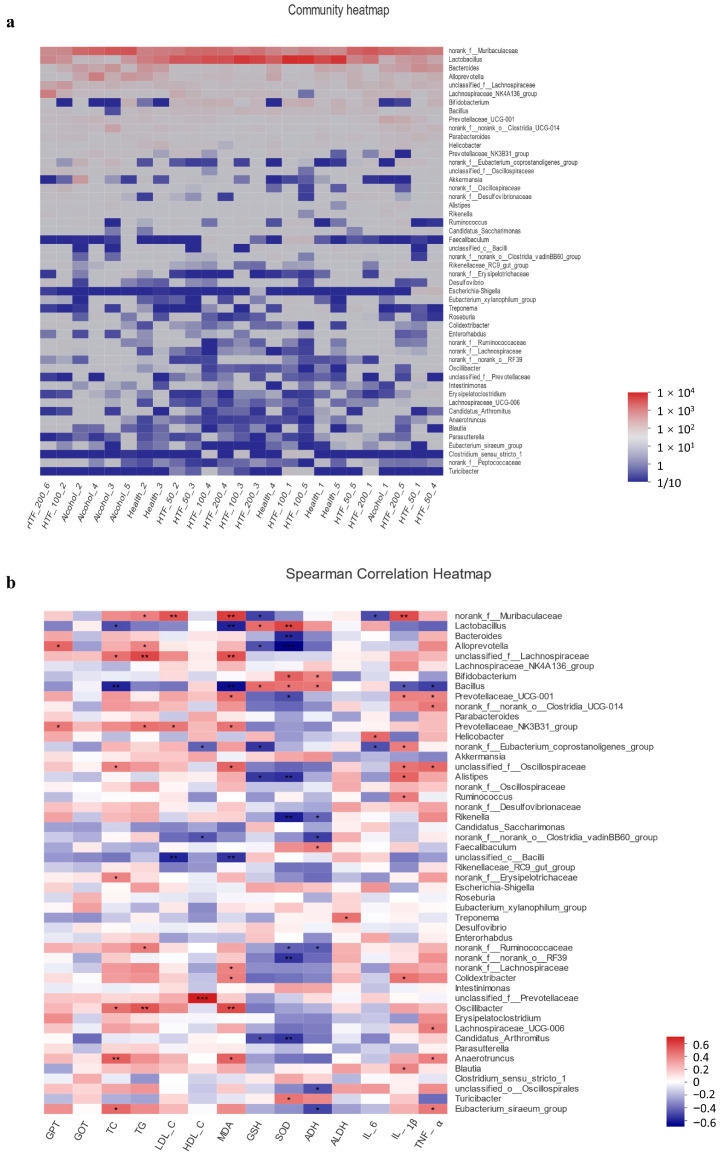
Correlation analysis. (**a**) Community heatmap analysis at the genus level. The color was defined at the phylum level. The *x*-axis was the sample name. (**b**) Pearson correlation between gut microbiota and metabolic factors at the genus level. * *p* < 0.05, ** *p* < 0.01, and *** *p* < 0.005.

**Table 1 nutrients-14-03662-t001:** Compounds in HTF which Related to Hepatoprotection, their content and biological effects.

CAS	Components	Content (%)	Outcomes and Potential Molecular Mechanisms	Biological Activities	OB (%)	DL	Contribution (mg)	Ref.
22255-13-6	Quercetin-3-O-arabinoside (Guaijaverin) *	2.72	Reduction in the levels of IL-1β, IL-18, and Caspase-1 inhibits the expression of P62 and Pink1; inhibition of MAPK and PI3K/Akt signaling pathways	Antioxidant and anti-inflammatory	29.65	0.7	0.016	[24,25]
23627-87-4	Kaempferol-3-O-galactoside (Trifolin)	2.05	Activation of aldehyde dehydrogenase; radical-scavenging activity	Antioxidant and hepatoprotective activity	19.61	0.74	0.008	[26]
490-46-0	Epicatechin	1.16	Amelioration of high circulating levels of lipids and endotoxins, and mitigates systemic inflammation; ease Hepatic dysregulation of lipid metabolism; inhibition of SCAP and prevents the activation of SREBP-1c	Ease hepatic dysregulation of lipid metabolism	28.93	0.24	0.007	[27]
117-39-5	Quercetin	0.51	Reduction in the levels of TNF-α; inhibition of the lipoxygenase and cyclooxygenase pathways	Anti-inflammatory	46.43	0.28	0.005	[28]
20315-25-7	Procyanidin B2	0.31	Proliferation inhibited and apoptosis induced in HSCs; down-regulate the expressions of VEGF-A, HIF-1α, α-SMA, Col-1 and TGF-β1 of HSCs	Hepatoprotective effect and anti-inflammatory	67.87	0.66	0.004	[29]
480-18-2	Dihydroquercetin (Taxifolin)	0.34	Inhibit the expression of P2X7R, IL-1β, and caspase-1; exhibit an inhibitory effect on lipid accumulation	Hepatoprotective effect and anti-inflammatory	57.84	0.27	0.004	[30]
5373-11-5	Luteolin-7-O-glucoside (Cynaroside)	2.60	Inhibited HMGB1/TLR4/NF-κB/MAPKs signaling pathways	Antioxidant	7.29	0.78	0.004	[31]
154-23-4	Catechin	0.46	Superoxide anion and superoxide-scavenging activity; suppress inflammation-related signal expression, including TNFA, COX-2, and iNOS	Antioxidant and anti-inflammatory	29.86	0.02	0.003	[28]
549-32-6	Quercetin-3-O-xyloside (Reynoutrin)	5.06	Inhibit the transcriptional activity of nuclear factor kappa-B	Antioxidant	1.68	0.7	0.002	[32]
17650-84-9	Kaempferol-3-O-rutinoside (Nicotiflorin)	2.08	Reduce the levels of IL-1β, IL-6, TNF-α, IFN-γ; decreased the MDA levels; increase GSH and the SOD activity; decrease the AST, ALT level	Hepatoprotective effect	3.64	0.73	0.002	[33]
572-30-5	Avicularin (Quercetin-3-O-α-L-arabinofuranoside)	3.47	A high radical-scavenging activity	Hepatoprotective effect	2.06	0.7	0.001	[34]
16290-07-6	Kaempferol-7-O-glucoside	2.21	NF-κB inhibitor	Antioxidant, anti-inflammatory and hepatoprotective activity	41.88	0.24	0.001	[35]
520-18-3	Kaempferol (3,5,7,4′-Tetrahydroxyflavone)	0.14	NF-κB inhibitor	Antioxidant, anti-inflammatory and hepatoprotective activity	41.88	0.24	0.001	[35]
480-41-1	Naringenin (5,7,4′-Trihydroxyflavanone)	0.10	Decreased levels of plasma and tissue total cholesterol; inhibition of oxidative stress through TGF-β pathway and prevention of the trans-differentiation of hepatic stellate cells (HSC). Pro-apoptotic effect, inhibition of MAPK, TLR, VEGF, and TGF-β, modulation of lipids and cholesterol synthesis, triglycerides, free fatty acids, HMG CoA reductase and collagen content	Hepatoprotective effect and anti-inflammatory	42.36	0.21	0.001	[36,37,38]
491-50-9	Quercetin-7-O-glucoside	1.31	Reduction in the levels of TNF-α, inhibition of COX2 and iNOS protein expression, inhibition of cow milk xanthine oxidase	Hepatoprotective effect	2.85	0.79	0.001	[28]
480-20-6	Aromadendrin (Dihydrokaempferol)	0.11	Regulation of the Keap1/Nrf2 pathway and regulate oxidative stress	Ameliorates severe acute pancreatitis	24.15	0.24	0.001	[39]
99882-10-7	Kaempferol-3-O-arabinoside	1.14	Antioxidant and anti-apoptotic properties, increase GSH and the SOD activity, decrease the AST, ALT level	Hepatoprotective effect	2.73	0.65	0.001	[40]
19833-12-6	Myricetin-3-O-glucoside	1.38	increase GSH and the SOD activity, decrease the AST, ALT level	Hepatoprotective effect	1.43	0.79	0.000	[41]
28608-75-5	Luteolin-8-C-glucoside (Orientin)	0.87	Inhibition of LPS-induced hyperpermeability in HUVEC cells	Anti-inflammatory	1.79	0.75	0.000	[42]
571-74-4	Sexangularetin	0.02	Decrease in the inflammatory markers IL-1β and myeloperoxidase	Anti-inflammatory	62.86	0.3	0.000	[43]
153-18-4	Rutin	0.11	Lower triglyceride content and abundance of lipid droplets; reduce cellular malondialdehyde level and restore superoxide dismutase activity in hepatocytes; suppress TGF-β/Smad signaling pathway	Hepatoprotective effect; dysfunctions of lipid metabolism	3.2	0.68	0.000	[44,45]
520-36-5	Apigenin	0.01	Inhibition of PI3K/Akt/mTOR pathway; activate the SIRT1 pathway; inhibit hepatic stellate cell activation and autophagy via TGF-β 1/Smad3 and p38/PPAR α Pathways	Hepatoprotective effect	23.06	0.21	0.000	[46]
124027-51-6	Quercetin-3-O-(6″-acetyl)galactoside	1.89	NI	NI	NI	NI	-	[47]
52525-35-6	Quercetin-3-O-robinobioside	1.00	Reactive oxygen species scavenging activity	Hepatoprotective effect	NI	NI	-	[48]
47705-70-4	Cyanidin-3-O-glucoside (Kuromanin)	0.52	Reduction in the levels of IL-1β, IL-6; activate mitophagy via the PINK1-PARKIN signaling pathway	Anti-inflammatory	NI	NI	-	[49,50]

OB, oral bioavailability; DL, drug-likeness; NI: no information has been found; *, we speculated it might to be the key substance which could alleviate acute liver damage.

## Data Availability

All study data are included in the article and Appendix A. All microbial diversity raw data has been uploaded to National Center for Biotechnology Information (NCBI) Sequence Read Archive (SRA) database (BioProject Number: PRJNA853465). Other data supporting the findings of this study are available from the corresponding authors upon reasonable request.

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
