# Peer review of "Hawk Tea Flavonoids as Natural Hepatoprotective Agents Alleviate Acute Liver Damage by Reshaping the Intestinal Microbiota and Modulating the Nrf2 and NF-κB Signaling Pathways"

_nutrients, 2022, doi:10.3390/nu14173662_

Round 1

Reviewer 1 Report

This manuscript is interesting and generally well written. However, it presents some flaws that must be resolved. In particular: 

Introduction: since authors studied the NRF2/KEAP1 pathway, this pathway deserves to be introduced. In fact, it deserves to be specified that this pathway not only preserves cells from oxidative stress but plays also a key role in cancer progression and chemotherapy resistance in many types of cancer (as also recently reviewed PMID: 35901941, 34070502, 33805996)

Acronyms must be written in full length when mentioned for the first time

2.8 ELISA Kit Analysis: ELISA product codes must be reported

2.9 Western Blot Analysis: the product codes and dilutions of primary antibodies must be reported

Figure 2: Figure is unreadable. Improve image quality or divide it in two figures. (k) authors must add the molecular weights in western blot image and specify what treatment has been loaded in each lane. Blots quantification analysis of each protein must be shown.

Figure 4 and 5: Figure is unreadable. Improve image quality

Lines 491-494: This paragraph should be moved in the introduction and integrated highlighting the multifaceted role of this pathway since, in the introduction, the Nrf2/keap1 pathway is only mentioned. 

Line 498: authors should underline that due to the key role of Nrf2 in inducing an antioxidant response, several drugs that are Nrf2 inducers are under evaluation for clinical applications (PMID: 33123312).

Reviewer 2 Report

The overall topic of the paper is of interest as more and more research tends to find natural solutions to treat diseases. It is also of interest to valorize traditional plants which are locally used and known for their benefits but for which the mechanisms of action are not yet elucidated.

The huge amount of experimental work which has been done and reported by the authors should be recognized. But this amount of data is not enough valorized because sometimes not readable. Indeed, several figures are too small for the reader to be able to get all the information.

Moreover, all the part related to extract compounds identification and evaluation of content, which is of high importance for this research paper, is not enough described.

 Following points are some comments and recommendations to the authors.

The material and methods section miss a part describing the chemicals/reagents used and their source.

Figure 1a is a visual description of the process used to prepare the extract: should “powder” be written instead of pounder?

Figure 1b: could the authors explain what HTF “before” and “after purification" means? It is not clearly explained, neither in the material and methods nor in the results section.

Figure 2a is too small to be read. Same comment for Figures 4b, 4f, 4g, 4h, 5 and all S1. Are all the figures necessary in the core of the paper or could they be added to supplemental figures to enlarge them?

Section 2.2 Total Flavonoids Quantification:  this paragraph describes the total flavonoid quantification. However, no data related to this measurement seems to be shown in the results section. Or if it is, it is not evidently shown.

Paragraph 3.1 of the results section and data related to the characterization of the HTF:

This part of the work is key for the paper as it corresponds to the characterization of the extract later tested in vivo for its activity. The authors also draw conclusion on the potential active compound based on this part of work. A lot of important information is missing regarding this characterization. A description of the approach applied for the identification of the different compounds (use of chemical standards, fragmentation, comparison with MS spectrum in databases,…) is not reported. The authors did a huge amount of work for this part of the paper, but approach and data are minimized. This part should be better presented and could even be one paper by itself if fully described.

A general comment on the Table S1 is that conclusions are done from the estimation of compound content and a ranking is established for each compound. This is used by the authors to estimate the potential responsibility of each compound on the health effect observed in vivo, but this estimation of content is not accurate analytically speaking. Indeed, what is the unit of the content column? Could the authors explain how this content has been determined? Is it the MS response, peak area (as deduced from the scale “.107”)? If yes, I would recommend not ranking the content of the compounds in the extract with this information. A MS response/signal is not proportional to the concentration of the injected compound. Molecules ionize differently depending on their chemical structure and their environment (matrix effects, solvent,…). To perform an accurate quantification, chemical standards are necessary and used to tune the MS system to optimize the quantification, and to build calibration curves. I would recommend to better explain how the estimation of content was performed because leads the reader to wrong interpretation.

In table S1, could the authors explain what “*” (which appears after some compounds) refers to?

In table S1, how is the “content %” determined? The reader may understand that the sum of the individual contents has been done and each compound content expressed relatively to this sum. Could the authors describe how they obtained it?

In table S1, could the authors explain how the oral bioavailability (OB) and drug-likeness (DL) were determined/estimated. Were these parameters found from the literature? Were they calculated?

 - Line 64: DSS should be written in full letters as Dextran Sulfate Sodium (DSS).

- Line 112 to 114: indirect style should be used instead of direct style (for example “were dissolved” instead of “dissolve”).

- Lines 118, 119: UPLC is a trademark of Waters. UHPLC should be written instead of UPLC, especially because the equipment is not a Waters' one.

- Line 118 to 147: description of the LC-MS method and equipment used. The same information is repeated such as the volume of injection (Lines 120 and 130) and the type of MS equipment (Lines 120 and 134).

- Line 152: “VC”, the positive control, should be written in full letters in the text for the reader to understand (VC for vitamin C).

- Line 373: if the HTF treatment has been given to the mice before alcohol challenge, shouldn’t the authors say “prevented” rather than “restored”, which would have been the case if HTF had been given after the alcohol treatment and the development of the model? Indeed, according to the protocol, alcohol is given the same day 30 min after HTF treatment. So it should be said that HTF prevents rather than treat the effect of alcohol. This is what the authors say in the conclusion on line 683 as “potential protective natural compound”.  

- Line 408 to 409: this sentence is difficult to be understood. Could the authors rephrase to be sure that the reader get the information?

- Line 446 to 449: “we found that, as a flavonoid compound, the HTF can be metabolized….”. The data of the present paper do not show the metabolism of the different flavonoids in HTF. I would recommend that the authors rephrase this sentence to be sure the reader get the proper message.

- Line 647-648: “the overgrowth harmful bacterial entered the liver through the enterohepatic circulation”. The sentence, as it is written, is understood as “the bacteria have invaded the liver by the enterohepatic circulation”. Is it what the authors wanted to say?

Round 2

Reviewer 1 Report

The manuscript has been significantly improved and can be accepted in the present form.

Reviewer 2 Report

I thank the authors for having carefully answered the questions and addressed the comments made.

Line 183: should be written "UHPLC" and not "HUPLC".

Line 608-610: HTF is rather a mixture of flavonoids than a flavonoid.